# Determination of Waste Industrial Dust Safety Characteristics

**DOI:** 10.3390/ijerph16122103

**Published:** 2019-06-14

**Authors:** Ivana Tureková, Eva Mračková, Iveta Marková

**Affiliations:** 1Department of Technology and Information Technologies, Constantine the Philosopher University in Nitra, Tr. A. Hlinku 1, 949 74 Nitra, Slovakia; ivaturekova@gmail.com; 2Department of Fire Protection, Technical University in Zvolen, Masaryka 24, 960 53 Zvolen, Slovakia; mrackova@tuzvo.sk; 3Department of Fire Engineering, Faculty of Security Engineering, University of Žilina, Univerzitná 1, 010 26 Žilina, Slovakia

**Keywords:** dust clouds, plastic, hazard, safety characteristics

## Abstract

This article deals with the assessment of the hazards of dust waste generated by modern CNC (computer numerical control) technologies from the processing of resin-containing plastic composites. The change of the original material into dusty waste predicts the emergence of new hazardous characteristics such as flammability, explosiveness and adverse effects on employee health. The aim of this article is to determine the experimental measurement of dust particle size, its thermal degradation and safety characteristics. Sieve analysis showed that the representative sample contained a 93.8% weight of particles with a size of less than 0.4 mm. Three degrees of thermal degradation of industrial dust samples and heat production (exo reaction ∑ΔH = 9172.9 J/g) were determined by TG (thermogravimetry), DTA (differential thermal analysis) and DSC (differential scanning calorimetr) methods. The measurement safety characteristics such as the lower explosion limit, the maximum explosion pressure, the maximum pressure rise rate, and the calculated cubic constant confirmed that the dust is an explosive, and is determined as explosion class St1 (determined by Cubic constant).

## 1. Introduction

Industry 4.0 is commonly referred to as the fourth industrial revolution [1], which also significantly relates to the Slovak automotive industry. This is the name for the current trend of automation and data exchange in manufacturing technologies. It includes cyber-physical systems, the Internet of things and cognitive computing [2,3,4,5]. New technologies bring products and wastes requiring knowledge of their hazardous features. Abrasive processes from plastics and composites due to CNC (computer numerical control) machinery leads to the occurrence of dust matter which might be explosive [6,7]. Dust is the most common injurant in which humans are exposed to during their work activities. The scope of harmful effects of dust on humans is very broad [8]. Their assessment depends on the origin, characteristics and size of the dust particles, its concentration in the air, the length and conditions of its action and the individual’s sensitivity to dust [9,10]. Therefore, it is an essential prerequisite to know the materials used for work as well as the physical and chemical properties along with safety characteristics which also predict health and environmental properties of waste [11].

### 1.1. Industrial Dust and Its Risk

From a health point of view, industrial dust is a polydisperse solid aerosol created by human activity during the mechanical processing of solid materials [12]. The size of dust particles is 1 to 100 μm, the particles larger than 30 μm are referred to as coarse dust and they sediment rapidly. The size of particles is determined by the actual or aerodynamic diameter. Particles larger than 10 μm settle in close proximity to the source site after a few minutes of their origin (or re-emergence). Therefore, particles smaller than 10 μm predominate in contaminated air. Particles with the size of 1μm settle only very slowly and particles smaller than 0.1 μm almost do not settle at all [13,14].

Depending on the external conditions, dusts act bimodally, they can burn and they may also have explosive properties in the whirled state [15]. Dust explosions in the industry might arise from many sources of initiation including friction and burning or smouldering materials [16].

In general, all types of dust are harmful and can cause serious health problems. Particles smaller than 10 μm are especially dangerous for humans as they can penetrate deep into the airways, and by inhaling air they get into the lung alveoli. For measurements in practice, it is functional to divide dust into two size fractions (i.e., components)—respirable and non-respirable [17,18].

While the larger particles (above 10 μm) can only cause upper respiratory irritation with coughing and sneezing and irritation of ocular conjunctivas, smaller particles can enter lower airways and particles of a size < 2.5 μm can permeate into lung passages and either settle in lungs or penetrate into the blood stream. Based on this aspect, the dust indicator can be divided into [11,19] as—Total Suspended Particle—TSP; Particles smaller than 10 μm (Particle Matter—PM_10_) and Particles smaller than 2.5 μm (PM_2.5_).

In terms of explosion hazard, combustible dust is comprised of small solid particles in the air that settle down by their own weight or might remain suspended in the air for some time. In general, particle size is defined by two parameters (which are significantly larger than the third) smaller than 500 μm [20].

With particles of a mean size larger than 400 μm, most dusts can no longer be initiated by standard energy. However, with an addition of 5 to 10% by weight of fine dust particles with a mean particle size of approximately 40 μm the mixture can again become explosive. It is important to bear in mind that when handling dust, larger particles become smaller as a result of abrasion [21].

### 1.2. Characteristic of Dust Risk Management

Risk management is a decision-making process that is based on the results of the risk assessment and aims to reduce risk while [22]:An objective assessment of the safety risk of dust explosion is a thorough method encompassing identification of all hazards, the probability of their occurrence and severity of their potential consequences.The health risk assessment is a process of assessing the probability and severity of a harmful effect on humans resulting from an exposure to a risk factor under defined conditions and from defined sources of risk consisting of hazard identification, exposure assessment, dose and effect relationship assessment and the characterization of the risk assessment uncertainties.

The first step is to determine the predisposition of equipment and technology to explosion of the equipment and consequently, take precautions to avoid the explosion (Figure 1) [22].

Depending on the level of danger that potential explosions threaten, additional steps may be needed to safeguard nearby equipment and personnel from unlikely, yet calamitous, explosions that transpire in spite of preventive techniques [23].

The dust and its suspending atmosphere must exhibit several characteristics. If any of the fire conditions (dispersed dust, sufficient air-oxygen and a sufficiently strong ignition source) is not present, a dust-cloud explosion cannot occur [20,22].

The aim of the paper is to determine the experimental determination of particle size, its thermal degradation and safety characteristics.

## 2. Materials and Methods

The industrial dust sample was made of plastic in combination with the resin. As in the industrial environment, a representative sample was taken from three sites on a CNC 5-axis MX 5 machining station for composites. With that, 100 g of sample from each sampling site was mixed and used for screen analysis. The machining area was provided by continuous suction into one central suction system.

### 2.1. Sieve Analysis of Dust Sample

The sieve analysis is a fractional or separation technique of particle size analysis. It is based on the use of a set of sieves with a particular size of openings which is assembled in the direction of gravitational transfer of the analyzed matter into the block with a gradually decreasing size of openings. After completion of fractionation, each sieve contains a particular portion of the original sample within the ranges determined by the size of the upper and lower sieve openings. The residues on the sieves are then weighed and evaluated. The sieve analysis was carried out on a RETSCH AS 200 (Laboratory of Technical University in Zvolen, Zvolen, Slovakia) in accordance with ISO 3310-1: 2007-03 [24]. Coarse impurities were removed by a fraction separation of <0.400 mm, using a sieve set of openings between 0.040 mm and 0.400 mm. Once sieve analysis was completed, the analytical balance (KERN PLT 450-3M) was used to measure the weight of each sieve fraction which gave us information about the percentages of the particles present in the analyzed sample [25].

### 2.2. Thermal Analysis of Dust Sample

Thermogravimetry (TG) and differential scanning calorimetry (DSC) were used. These are analytical methods, as the weight of analyzed samples was in milligrams. These methods find application in observations and comparisons of the thermal decomposition of matter and in the observation of the changes of conditions of the course of chemical reactions. Thermogravimetry (TG) studies the course of thermolysis and polymer burning and records the changes in the weight of the heated sample. The samples (with an average weight of 10.100 mg) were stabilized for 24 h under standard conditions; The test was carried out on a Mettler TA 3000 (Fire Technical and Expert Institute, Bratislava, Slovakia) with a TC 10A processor and a TG 50 thermogravimetric weights module in the air with a flow rate of 200 mL∙min^−1^ and a platinum open crucible.

Differential scanning calorimetry provides information on endothermic and exothermic processes during the controlled thermal decomposition. By means of differential scanning calorimetry, the reaction heat dissolution temperature of the test sample was identified in the temperature interval of 35–600 °C in a dynamic air atmosphere. The samples (with an average weight of 4.50 mg) were analyzed by a DSC module (Fire Technical and Expert Institute, Bratislava, Slovakia) with full DSC 20 and Grapheware TA-72.2/5 evaluation software. Two replicates (separated fraction < 0.400 mm) were tested for each sample.

### 2.3. Combustible Dust Characteristics

With regard to fire and explosion prevention, the following parameters can be considered as dust safety characteristics [11]:Lower explosive limit (LEL)Maximum explosion pressure p_max_Maximum rate of pressure rise (dp/dt)_max_Cubic constant K_St_

The cubic constant allows assessment of the effects of the explosion in the given closed vessel and is the basis for classification of dust explosion according to classes St1, St2 and St3 (Table 1).

The K_st_ value is calculated from the Cubic law according to a defined relationship between (dp/dt)_max_ and the volume (V) in which the explosion occurs:K_st_ = (dp/dt)_max_ × V^1/3^(1)
where is K_st_—cubic constant for the mixture of dust and air (bar/m·s); (dp/dt)_max_—maximum rate of pressure rise (bar/s); V—volume of the cubic container (m^3^).

Evaluation of these characteristics, with the exception of the cubic constant, is not yet possible; therefore the most reliable results are determined experimentally as well as for the needs of the practice, in the accredited laboratory. If the matter composition is known, we can roughly determine these characteristics, for instance from tables. In real practice, in cases of new materials and their changes, these characteristics must be determined to ensure the correct design of the suction equipment [27].

## 3. Results

### 3.1. Results of the Sieve Analysis

The results of the sieve analysis of the dust particles smaller than >0.400 mm show that their weight representation is 93.8%. The average size of dust subjected to the sieve analysis was 0.049 ± 0.01 mm (Figure 2). This uncertainty is standard expanded uncertainty, corresponding to a 95% confidence level.

### 3.2. Results of Thermal Analysis—Thermogravimetry (TG)

The thermogravimetric analysis was performed in the heat interval of 35–600 °C in the air dynamic atmosphere (for the purpose of the simulated real situation). The individual degrees of thermal decomposition and the corresponding residuals were defined. The results of the measurements are shown in Figure 3 and Table 2.

The thermal decomposition of the industrial dust took place in three stages. The first stage of thermal decomposition (40–70 °C) can be attributed to the oxidative reactions of the resin [28]. The active dust decomposition phase was conducted at 510 °C. Thus, it can be noted that the amount of dust sample that was not subject to degradation in the second stage of thermal decomposition was decomposed in the third stage at much higher temperatures where the highest weight loss was recorded. The experimental sample was a macromolecular substance. Its thermal degradation is similar to other organic material [29].

### 3.3. Results of Thermal Analysis—DSC Results

The DSC method was used to measure changes in the reaction enthalpy of the fuel forming and heat in the chosen interval of exothermic reactions. The reaction heat was determined and the maximum heat generation rate was characterized by the maximum exothermic peak temperature on the thermal analytic curve in the temperature range of 35–600 °C in a dynamic air atmosphere. The first step identified in TG, DTA at 40–70 °C was not confirmed in DSC analysis. It is possible to assume the consumption of released heat for the oxidation reactions of the components in the dust (Table 3).

The first changes in thermal stability of industrial dust are monitored by DSC at 190 °C. In the chosen interval of exothermic reactions, the reaction heat was determined and the maximum heat generation rate was characterized by the maximum exothermic peak temperature on the thermal analytic curve in the temperature range of 35–600 °C in a dynamic air atmosphere (Table 3).

By the use of DSC, temperature intervals of the exothermic reaction were observed, the course of the reaction enthalpy during heating without endothermic reactions is shown in Figure 4.

The reaction enthalpy changes determined by this means are not consistent with enthalpy measurements in a calorimetric bomb, but provide substantial information about the heat release in thermal degradation processes and allow us to collect information about the thermal color reactions in the individual degrees of thermal decomposition of the test sample.

### 3.4. Results of Safety Characteristics

Results of safety characteristics were realized in two steps. The first step was experimental measurements of p_max_ (Figure 5), LEL (Figure 6) and (dp/dt)_max_ in Explosion Cubicle VA-20L with Sartorius weight by EN ISO/IEC 17025:2005 (Figure 5) [30]. Ignition strength was 2 kJ.

The second step was the calculation of K_st_ by Equation (1). The combustible dust characteristics of industrial dust are shown in Table 4. In terms of explosion classification, the industrial dust sample is classified as explosion class 1.

## 4. Discussion

The average particle size of 0.049 mm points out the fact that particles can settle, but it is important to ensure that there is no excessive accumulation of a dust pile and no explosive concentration is reached, especially by primary or secondary dust leakage from equipment.

Primary dust leakage can be caused by the opening of equipment which is part of a routine technological process, i.e., opening due to filling or removing of the material but also from all open places from which dust can escape the equipment. Premises of CNC machinery and storage containers of filter units are the most critical.

Secondary leakages, on the other hand, represent occasional openings for the purpose of taking samples, control, e.g., changing of the filter cartridges on filter units, maintenance and repairs of equipment in case of breakdowns, cleaning and control openings in pipelines.

The results acquired by testing should be implemented in the form of technical measures using engineering instruments into real technology or supplemented with further testing:

- Provide sufficient suction from the grinding process.

- Prevent the formation of continuous layers of dust with thickness greater than 1 mm by a suitable cleaning regime.

- The operational filter unit must be designed for suction of combustible dusts and dimensioned to the explosion characteristics of the collected dust.

- The filter unit and piping must withstand a maximum explosion pressure in accordance with EN 14,460 [31], alternatively, explosion-proofing equipment or explosion suppression equipment must be installed.

- Separating explosions on the supply line from the filters must be ensured (to prevent the transmission of an explosion through the pipeline to operator crew).

- Develop an explosion protocol.

The results from TG and DSC methods demonstrate that industrial dust releases sufficient energy and is, therefore, a suitable waste to be used for energy recovery by combustion. However, it is important to acquire enough information about the actual amount of generated waste, the exact chemical composition of the product of thermal combustion (including dust captured from the filter) and propose the mode of transport, i.e., pre-treatment by, e.g., tableting.

Dustiness in the workplace, caused by dust generation, has detrimental effects on the human organism [32] and represents one of the basic problems in the field of occupational safety and health at the workplace. In terms of health risk, a more detailed analysis of the operation with the aim of particle PM_2.5_ and PM_10_ particle capture in the operator’s respiratory zone and to assess the length of exposure at the workplace.

## 5. Conclusions

This article pointed to the fact that new technologies can bring new risks, be they in safety, the environment or in health. The creation of new components is associated with waste generation, which is classified as waste, with a catalog number. Dust can cause an explosive atmosphere, and in the form of inhalable fractions can also have a negative impact on human health in the case of long-term exposure depending on the size of the dust particles (93.8% by weight of particles smaller than 0.4 mm). In our study, more than half of experimental industrial dust samples passed through a 50 micron sieve.

TG results and DTA analysis identified the first heat degradation changes in the dust sample (0.4 mm fraction) in the 40–70 °C temperature range. The temperature of 40 °C is a low temperature that can be reached in the working environment in extreme conditions. The main thermal decomposition of the composite dust was in the interval of 407–584 °C with a high resistance residue which declares relative thermal stability. DSC analysis confirmed the dependence of the release of the released reaction rate, where a 190 °C temperature is required.

The lower explosion limit was determined at an energy initiation of 20 g/m^3^ and the dust was included in the St1 explosion class. These data are important for designing measures for employees working in explosive environments—ATEX Directive 137 and equipment and protection systems could be introduced into an environment with a danger of explosion ATEX 100. A confirmed dust explosion will require explosion protection documentation and a classification of the environment of the explosion hazard. The suction device—filters are the most critical element from this technology. Filters are cleaned with air pulses and this area is classified as dangerous from the point of flammable dust explosion (Zone 20).

## Figures and Tables

**Figure 1 ijerph-16-02103-f001:**
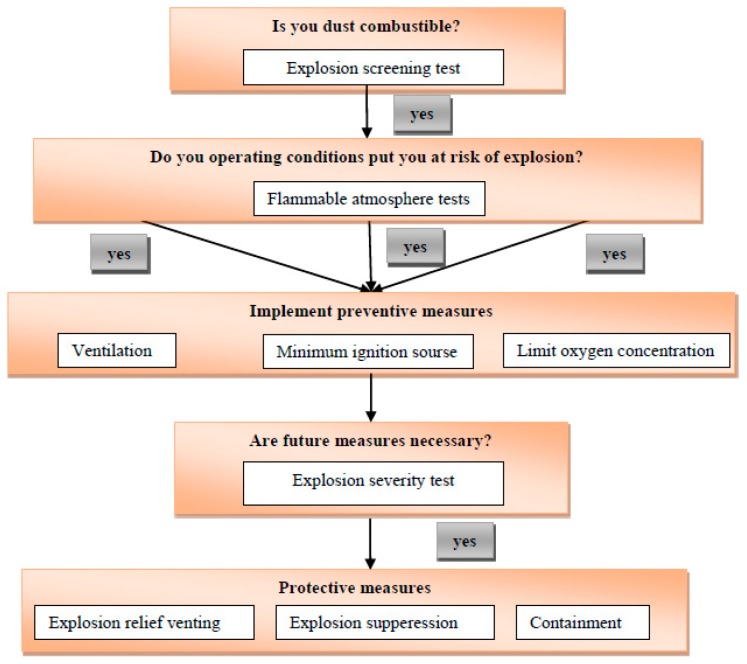
Procedure for applying preventive measures [22].

**Figure 2 ijerph-16-02103-f002:**
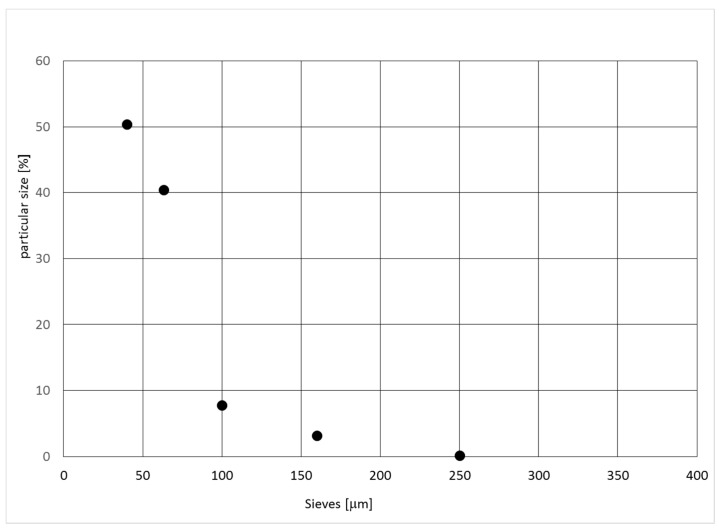
Particle size in % of the industrial dust.

**Figure 3 ijerph-16-02103-f003:**
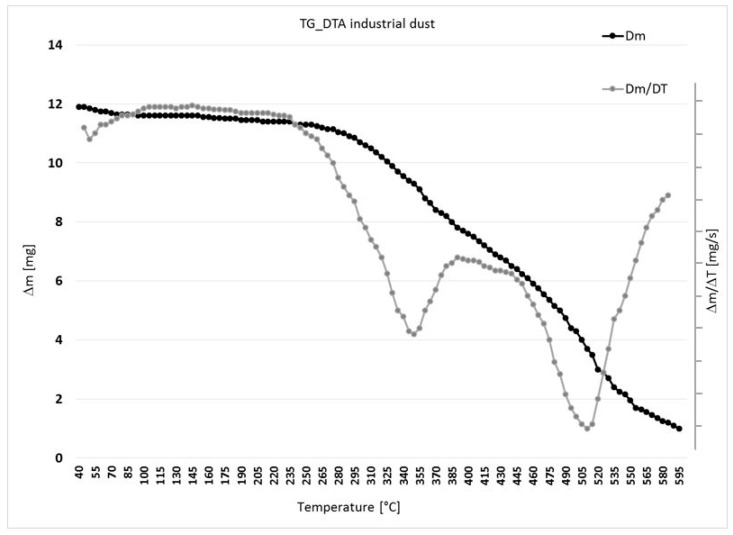
TG a DTA analysis of the industrial dust.

**Figure 4 ijerph-16-02103-f004:**
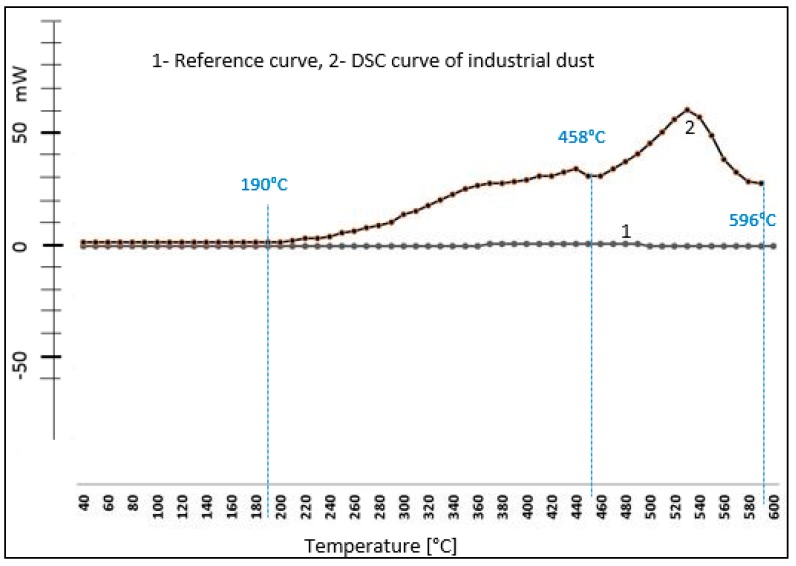
Description of what is differential scanning calorimetry (DSC) analysis of industrial dust.

**Figure 5 ijerph-16-02103-f005:**
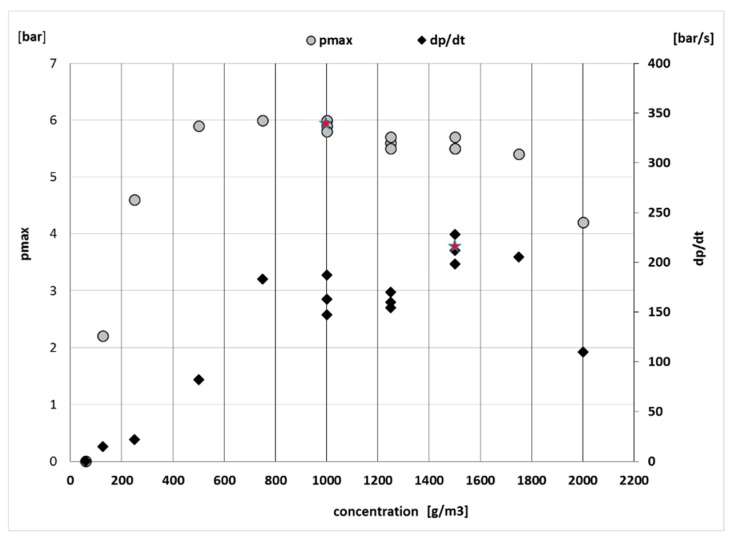
Influence p_max_ on concentration of industrial dust.

**Figure 6 ijerph-16-02103-f006:**
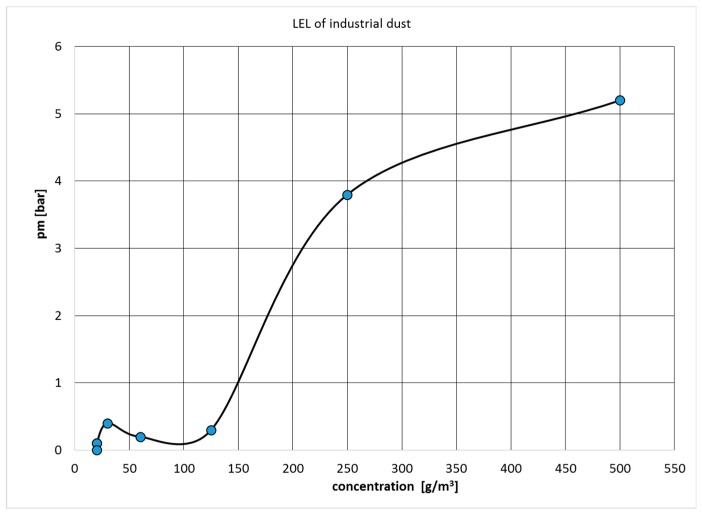
Determinate of industrial dust´s lower explosive limit (LEL).

**Table 1 ijerph-16-02103-t001:** The industrial dust explosion classes based on K_st_ value [26].

Explosion Class	K_st_ (bar/m·s)
St1	≤200
St2	200–300
St3	>300

**Table 2 ijerph-16-02103-t002:** Temperature characterization of individual degrees of decomposition of industrial dust sample by thermogravimetric (TG) analysis.

Sample	Degrees of Decomposition	Heat Interval (°C)	Weight Loss (%)	Tmax (°C)	Resistant C (%)
Industrial dust	I.	40–70	1.67	51	98.33
		139–407	22.40	350	75.93
	II.	407–584	34.41	511	41.52

For temperature calibration, materials with given Curie temperatures were used: Isatherm (144.5 °C); Nickel (357.0 °C), a Trafoperm (748.0 °C); Accuracy of test equipment was determined from calibrations: (–2.4 to + 8.2%).

**Table 3 ijerph-16-02103-t003:** Dependence of changes in reaction enthalpy of industrial dust samples on temperature.

Sample	Degrees of Decomposition	Heat Interval (°C)	Change of Reaction Enthalpy (J/g)	Maximum Peak Temperature (°C)
Industrial dust	I.	190–458	3997.9 (exo)	435.9
	II.	458–596	5175.0 (exo)	41.52

**Table 4 ijerph-16-02103-t004:** The combustible dust characteristics of industrial dusts.

Type of Dust	LEL (g/m^3^)	p_max_ (bar)	(dp/dt)_max_ (bar/s)	K_st_(bar/m·s)	Explosion Class
Industrial dust	20	5.9	213	58.0	St 1

LEL: lower explosive limit.

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
