# Peer review of "Determination of Waste Industrial Dust Safety Characteristics"

_ijerph, 2019, doi:10.3390/ijerph16122103_

Round 1

Reviewer 1 Report

The article is well structured, brings original results of experimental work for dust safety characteristics. Maybe the introduction part is too broad, but it is good base for experimental part.

I have found only one small formal mistake – in line 123 please change 0.400 mm instead of 0,400.

I have recommended the article for publication.

Author Response

Thank you for your review.

Point 1: I have found only one small formal mistake – in line 123 please change 0.400 mm instead of 0,400.

Response 1: Mistake  in line 123 was correct.

Written modifications and additions on the reviewer ´s requirements are highlighted in pink.

Reviewer 2 Report

Manuscript Number: ijerph-505780

Title: Determination of Waste Industrial Dust Safety Characteristics

Summary of the article - Main impressions:

I would like to thank the authors for their work and effort in providing data on determination of waste industrial dust safety characteristics. The experimental approach presented in this work seems promising. The work could be of value to the fire community; however, the authors are encouraged to implement major revisions to the paper, by addressing the following topics, in order to improve its quality and make it suitable for publication. I hope the authors will be able to continue this interesting work and address the issues raised.

Comments:

The abstract needs further improvement.

p. 1 (Section 1): Section needs to be re-organized in order to clearly state that the aim of this work. Authors are encouraged to enhance the introduction section by reorganizing the first 2 paragraphs.

p. 2 (line 66):    authors are encouraged to provide more information regarding the definition of those two diameters referred to the text, maybe even add some references.

p. 2 (line 74):    authors are encouraged to enhance the quality of Figure 1.

p. 4 (line 127):  what type of crucibles did the authors used? Hoe many milligrams of samples? Did they use any lids? Did they account of any pressure effects? Did author performed repeatability tests?

p. 5 (line 174):  in Figure 3 could you please comment on the mass loss occurring in the region of 40-70 oC?

p. 6 (line 193):  in Figure 4 please adjust the scale and present the data of Specific Heat instead.

p. 7 (line 200):  please provide more information on how the material was classified.

Recommendation:

Minor Revision

Author Response

Thank you for the helpful comments we have incorporated into the article. We believe that editing the article has made the article better.

The abstract needs further improvement.

The abstract was revised

Point 1: p. 1 (Section 1): Section needs to be re-organized in order to clearly state that the aim of this work. Authors are encouraged to enhance the introduction section by reorganizing the first 2 paragraphs.

Response 1: Yes, we agree and repaced our the first 2 paragraphs.

Point 2: p. 2 (line 66):    authors are encouraged to provide more information regarding the definition of those two diameters referred to the text, maybe even add some references.

Response 2: Information has been added

Point 3: (line 74):    authors are encouraged to enhance the quality of Figure 1.

Response 3: The reservations in Fig. 4 had other opponents. We decided not to include this Fig. 4 in the article.

Point 4:  p. 4 (line 127):  what type of crucibles did the authors used? Hoe many milligrams of samples? Did they use any lids? Did they account of any pressure effects? Did author performed repeatability tests?

Response 4: Thank you for very important notice fot thermal analysis. We added it.

Point 5:  p5 (line 174):  in Figure 3 could you please comment on the mass loss occurring in the region of 40-70 °C?

Response 5: The answer is adedd in lines 190-202.

Point 6:  p. 6 (line 193):  in Figure 4 please adjust the scale and present the data of Specific Heat instead.p5

Response 6: p. 6. The Figure 4 is added about actual data. DSC analysis measurments Reaction Enthalpy. We thinks that this value is not Specific Heat. I future, we will measurment data from cone calorimether

Point 7:  p. 7 (line 200):  please provide more information on how the material was classified.

Response 7: More information are in chapter 3.4

Written modifications and additions to the requirements in the review are highlighted in green.

Reviewer 3 Report

The research mainly performs TG-DSC for industrial dust with the purpose of evaluating the effect of industrial dust on public safety and health. However, several major problem appears in the paper regarding both the logic of the content and the used techniques. 

The motivation of the research is not very clear. The logic connecting the TG results and safety evaluation has not been presented clearly. The authors mentioned risk management however it is not very clear how to use TG results to conducted risk management. 

The introduction is a bit of wordy. Please straight to the point. 

Why air? Why not perform nitrogen experiments? Nitrogen TG is the best way to identify the degrading mechanism of the material. The authors claimed two stages based on DTG however it seems there is another stage for 40 C. As presented in https://pubs.acs.org/doi/10.1021/ie402905z  resin will decompose under relatively low temperatures. The stage might be caused by resin oxidization. 

The two peaks behavior is typical polymer degradation. See https://doi.org/10.1016/j.combustflame.2006.04.013. We must understand more mechanisms before we judge a material. In fact, the decomposition temperature of 139 C, as claimed by the authors, is quite close to the one of wood, which supposedly should be relative safe materals.  

We usually use percentage to label TG results. 

Why is the heating rate 10 K/min? Why not perform multiple heating rate experiments? Why not keep chasing the kinetics. If the authors refer to https://pubs.acs.org/doi/abs/10.1021/ef501380c it can be easily found the merit of multiple heating rates. The behaviors will change. 

The base line of DSC should be drawn to identify the heat of pyrolysis of the materials. It is really weird the decomposition before 200 C has no dsc. So the TG results are caused by temperature shift of the balance? Besides, what do the two lines in Fig. 4 mean?

The conclusion is too general, which definitely need to be improved. 

Author Response

Thank you for the helpful comments we have incorporated into the article. We believe that editing the article has made the article better. We have made changes to the content the article. 

Written modifications and additions to the requirements in the review are highlighted in gray.

We have chosen following sequence of steps:

Brief introduction (line 103 is out), characteristic of industrial dust hazard, its network analysis. Selected experimental methods TG, DTA and DSC (the purpose is to evaluate the thermal stability of the material and the potential risk of a heat generation in exothermic reactions as a result of the released heat after sample decomposition). Explosion parameters were determined in the Explosion Chamber and the calculated dust inclusion value in Explosion Class. Our goal was to evaluate industrial dust as a flammable and explosive material. The work does not solve the risk management.

Thermal analysis was observed in the air atmosphere to simulate real conditions. Supplements listed

The article did not pay attention to the kinetics of events. In the future, we will consider the aforementioned option with the addition of nitrogen measurements.

Figure 4 has been modified for purposes of explaining ongoing events. TG, DTA experienced changes at 40-70 ° C. The thermal interval presents the course of the exo reaction as well as the endo and there is no change in the reaction enthalpy and the DSC curve did not. There was no weight shift during the experiment, and the experiments were repeated.

Reviewer 4 Report

The paper treats the new problem of the hazards of abrasive waste dust of plastic machining for the health of workers, polluting the environment and risk of explosion.

As authors mentioned, the smallest size dust is the most hazardous from an occupational point of view and require the least ignition energy. Now the very smal will only slowly settle or not at all. How do you know you took a representative sample?

Line 44. The heading text is not clear, what means 'dustind'?

Line 103: What do you mean with: "The particle-size distribution must be capable of supporting energy"?

Line 108: Same question: "industrial dust during it heat load for purpose to determinate safety parameter". It contains several language flaws.

Lines 192-193: The comma should be replaced by a semicolon.

Line 204: You have to report the experimental conditions under which the explosion indices data have been determined: what kind of vessel, ignition strength.

Line 223: 'pipping' is 'piping', I assume

Although you claim to have determined the hazard for the human worker, your reviewer cannot even find a dust particle distribution in your work. And what is the conclusion with respect to the TGA and DTA results in terms of hazard? And what do the explosion indices mean for the dust explosion hazard in more concrete, quantitative terms for the preventive and protective measures that shall have to be taken? Can you refer to the ATEX Directive?

Author Response

Thank you for your suggestions in the review. We have made changes to the content the article. 

Written modifications and additions to the requirements in the review are highlighted in blue.

We have chosen following sequence of steps:

Brief introduction (line 103 is out), characteristic of industrial dust hazard, its network analysis. Selected experimental methods TG, DTA and DSC (the purpose is to evaluate the thermal stability of the material and the potential risk of a heat generation in exothermic reactions as a result of the released heat after sample decomposition). Explosion parameters were determined in the Explosion Chamber and the calculated dust inclusion value in Explosion Class. Our goal was to evaluate industrial dust as a flammable and explosive material. The work does not solve the risk management only to characteriset.

Thermal analysis was observed in the air atmosphere to simulate real conditions. Supplements listed

The article did not pay attention to the kinetics of events. In the future, we will consider the aforementioned option with the addition of nitrogen measurements.

Figure 4 has been modified for purposes of explaining ongoing events. TG, DTA experienced changes at 40-70 ° C. The thermal interval presents the course of the exo reaction as well as the endo and there is no change in the reaction enthalpy and the DSC curve did not. There was no weight shift during the experiment, and the experiments were repeated.

Linia 192-193 was corrected.

More information about experimental conditions od explosion parameters are in chapter 3.4.

The conclusion is to supplement the ATEX information.

Round 2

Reviewer 3 Report

The revision can be accepted.

Author Response

We would like to thank the reviewer for the comments. Reviewer 4 also had concrete comments to English expressions and these were included in the text.

Reviewer 4 Report

Many changes after previous comments improved the paper considerably. Yet, some comments should still be made.

Line 22. Yes, the dust is explosive, but St1 is the lowest class of explosibility. It means that it will not come to explosion that easily and an explosion will not be as violent as that of a St3 class dust one, but it can still cause much damage. So, I would suggest to delete "very".

Line 33: '....with the help of CNC machinery ....' sounds as not good English. I would suggest: '....due to CNC machining....'

Line 64: What do you mean in this context with 'dimensions'? Are you thinking of parameters?

Line 104: 'following'? Do you mean 'dispersed dust', 'sufficient air-oxygen' and 'a sufficient strong ignition source'?

Line 146: noun 'measurement' should be a verb, e.g., 'analysed'.

Line 157: St3    30 should be 300.

Line 162: 'Determination'? And to which 'characteristics' do you refer, the ones on the previous page line 151-153? These can all be determined experimentally. The Cubic constant follows from (dp/dt)max at for explosion optimum conditions.

Line 175: It is 'particle size fraction in %...'

LIne 178: What do you mean with '... dynamic atmosphere of air to simulate real, the individual.....'? The wording is not understood and the sentences do not connect. Do you mean there was a flow of air?

Line 258: The sentence is not complete. Missing is behind analysis 'is required'.

Why don't you conclude that more than half of your dust sample passes a 50 micron sieve?

Author Response

Thank you for the helpful comments we have incorporated into the article. We believe that editing the article has made the article better.

Point 1: Line 22. Yes, the dust is explosive, but St1 is the lowest class of explosibility. It means that it will not come to explosion that easily and an explosion will not be as violent as that of a St3 class dust one, but it can still cause much damage. So, I would suggest to delete "very".

Response 1: Yes, thank you, very removed.

Point 2: Line 33: '....with the help of CNC machinery ....' sounds as not good English. I would suggest: '....due to CNC machining....

Response 2: Thank you, we fixed it.

Point 3: Line 64 What do you mean in this context with 'dimensions'? Are you thinking of parameters?

Response 3: Thank you, we fixed it.

Point 4:  Line 104: 'following'? Do you mean 'dispersed dust', 'sufficient air-oxygen' and 'a sufficient strong ignition source'?

Response 4: Thank you, we fixed it.

Point 5:  Line 146: noun 'measurement' should be a verb, e.g., 'analysed'.

Response 5: Thank you, we fixed it.

Point 6:  Line 157: St3    30 should be 300

Response 6: Thank you, we fixed it.

Point 7:  Line 162: 'Determination'? And to which 'characteristics' do you refer, the ones on the previous page line 151-153? These can all be determined experimentally. The Cubic constant follows from (dp/dt)max at for explosion optimum conditions.

Response 7:  We agree with the opponent's opinion. The term Determination did not express it exactly. The correct term is “Evaluation”. The Cubic constant follows from (dp / dt) optimum conditions, so we added experimental values dp / dt in figure 5.

Point 8:  Line 175: It is 'particle size fraction in %...

Response 8: The results of the sieve analysis of the dust particles smaller than < 0.400 mm show that their weight representation is 93.8 %.

We used dust samples smaller than< 0.400 mm for thermal analysis and added to the sentence: Two replicates (separated fraction <0.400 mm) were tested for each sample.

Point 9:  LIne 178: What do you mean with '... dynamic atmosphere of air to simulate real, the individual.....'? The wording is not understood and the sentences do not connect. Do you mean there was a flow of air?

Response 9: Thank you, we fixed it.

Point 10:  Line 258: The sentence is not complete. Missing is behind analysis 'is required'.

Response 10: Thank you, we fixed it.

Point 11:  Why don't you conclude that more than half of your dust sample passes a 50 micron sieve?

Response 11: Thank you, we added it in Conclusion.